# MicroRNA-100 Mediates Hydrogen Peroxide-Induced Apoptosis of Human Retinal Pigment Epithelium ARPE-19 Cells

**DOI:** 10.3390/ph14040314

**Published:** 2021-04-01

**Authors:** Yuh-Shin Chang, Yo-Chen Chang, Po-Han Chen, Chia-Yang Li, Wen-Chuan Wu, Ying-Hsien Kao

**Affiliations:** 1Chi Mei Medical Center, Department of Ophthalmology, Tainan 71004, Taiwan; yuhshinchang@yahoo.com.tw; 2Graduate Institute of Medical Science, College of Health Science, Chang Jung Christian University, Tainan 71101, Taiwan; 3Graduate Institute of Medicine, College of Medicine, Kaohsiung Medical University, Kaohsiung 80708, Taiwan; ycchang@kmu.edu.tw (Y.-C.C.); chiayangli@kmu.edu.tw (C.-Y.L.); 4Department of Ophthalmology, Kaohsiung Medical University Hospital, Kaohsiung Medical University, Kaohsiung 80708, Taiwan; 5Department of Ophthalmology, Kaohsiung Municipal Ta-Tung Hospital, Kaohsiung Medical University, Kaohsiung 80145, Taiwan; 6Department of Ophthalmology, School of Medicine, Kaohsiung Medical University, Kaohsiung 80708, Taiwan; 7Department of Medical Research, E-Da Hospital, Kaohsiung 82445, Taiwan; lion6151703@yahoo.com.tw; 8Department of Ophthalmology, China Medical University Hospital, Taichung 40402, Taiwan

**Keywords:** heme oxygenase-1, oxidative stress, microRNA biosynthesis, mTOR, signal transduction

## Abstract

This study investigated the regulatory role of microRNA 100 (miR-100) in hydrogen peroxide (H_2_O_2_)-induced apoptosis of human retinal pigment epithelial ARPE-19 cells. H_2_O_2_ induced oxidative cell death of cultured ARPE-19 cells was measured by cytotoxicity assay. qRT-PCR was used to quantify cytosolic and extracellular contents of miR-100. Kinase and miR-100 inhibition treatments were applied to determine the regulatory signaling pathways involved in cell death regulation. H_2_O_2_ dose-dependently reduced viability of ARPE-19 cells and simultaneously upregulated miR-100 levels in both cytosolic and extracellular compartments. Western blotting detection indicated that H_2_O_2_ elicited hyperphosphorylation of PI3K/Akt, ERK1/2, JNK, p38 MAPK, and p65 NF-κB. Further kinase inhibition experiments demonstrated that PI3K, p38 MAPK, and NF-κB activities were involved in oxidative-stress-induced miR-100 upregulation in ARPE-19 cells, while blockade of PI3K, JNK, and NF-κB signaling significantly attenuated the oxidative cell death. Intriguingly, MiR-100 antagomir treatment exerted a cytoprotective effect against the H_2_O_2_-induced oxidative cell death through attenuating the oxidation-induced AMPK hyperphosphorylation, restoring cellular mTOR and p62/SQSTM1 levels and upregulating heme oxygenase-1 expression. These findings support that miR-100 at least in part mediates H_2_O_2_-induced cell death of ARPE-19 cells and can be regarded as a preventive and therapeutic target for retinal degenerative disease.

## 1. Introduction

Age-related macular degeneration (AMD) is the most common cause of blindness among patients aged 65 years and older in the Western world [1], and incidence continues to rise due to the increasing percentage of older adults in the common population. In Asia, the prevalence of late AMD is estimated to be between 0.20 and 1.90%, based on data from China [2], Japan [3], Taiwan [4], and Singapore [5]. Pathologically, AMD results from retinal pigment epithelium (RPE) dysfunction or loss associated with photoreceptor fallout, Bruch’s membrane thickening, and choriocapillary hypoperfusion [6]. The RPE is a monolayer of pigmented cells forming part of the blood retina barrier and is particularly susceptible to oxidative stress due to the layer’s high consumption of oxygen. Etiology study findings indicate that AMD is a multifactorial late-onset disease, while oxidative-stress-induced RPE cell damage is suggested to be a critical factor [7,8,9,10,11,12,13]. Antioxidant supplementation is therefore believed to be a therapeutic strategy for AMD treatment through modulating redox status balance [7,9,10,11,14].

MicroRNAs (miRNAs) are a group of noncoding RNAs with 18~22 nucleotides in length that inhibit endogenous gene expression through cleavage of gene transcript through translational cleavage [15]. In both animal and human visual systems, miRNAs have been demonstrated to be involved in various aspects of retina development, including retinogenesis, retinal homeostasis, and retinal degeneration [16,17,18]. An increasing body of evidence also identifies miRNA signatures in vitreous humors and plasma of AMD patients and suggests that some candidate miRNAs are plausible therapeutic targets for gene therapy [19,20,21,22]. MiRNA-100 (miR-100) belongs to the miR-99 family, whose members include miR-99a, miR-99b, and miR-100, and has long been known as a key apoptotic regulator in various cell types, mostly in cancer cells [23,24,25,26,27,28,29,30]. Previous studies on cardiovascular system have suggested that elevated miR-100 is involved in heart failure [31], while miR-100 gene silencing protects oxidative-stress-induced apoptosis of cultured neonatal cardiomyocytes [32]. These findings support that miR-100 plays an important role in the regulation of oxidative-related necroptosis of ischemic cells. In the retina, hydrogen peroxide (H_2_O_2_)-induced oxidative stress has been demonstrated to specifically up-regulate miR-100 expression in cultured rat retinal ganglion RGC-5 cells undergoing apoptotic cell death, while miR-100 depletion exerts anti-apoptosis effect through insulin-like growth factor 1 receptor (IGF1R) and Akt/ERK/Trkβ pathways in the retinal ganglion cells [33]. MiR-100 is also found to be constitutively expressed in retinal microvascular endothelial cells, but its expression is suppressed by angiogenic stimulation during ocular angiogenesis [34]. Despite lacking clinical evidence, those findings strongly suggest that miR-100 is widely distributed in retinal tissues and contributes to ocular disorders. To date, the regulatory role of oxidative stress in miR-100 biosynthesis in RPE cells remains elusive. Despite lacking clinical evidence, we hypothesized that the oxidation-induced miR-100 upregulation in RPE cells is very likely involved in the pathogenesis of AMD. To test our hypothesis, this study aimed at characterizing whether H_2_O_2_-mimicked oxidative stress also induces ARPE-19 cell apoptosis via upregulating miR-100 biogenesis and its release, examining the signal pathways involved in the miR-100 upregulation, determining the cytoprotective effect of miR-100 antagomir treatment against oxidative cell death, and the possible underlying mechanism.

## 2. Results

### 2.1. Oxidative Stress Increased miR-100 Biosynthesis in Human ARPE-19 Cells

Recent studies have evidenced that the existence of miRNA in the extracellular vesicles released by RPE cells may contribute to retinal pathophysiolgical processes [35,36]. To characterize the profiles of miR-100 biosynthesis and its release in response to oxidative stress, we first examined the cytotoxic effect of H_2_O_2_-mimicked oxidative stress in human ARPE-19 RPE cells by exposing cells to various concentrations of H_2_O_2_, and consequent cell viability was determined by MTT cytotoxicity assay. The results showed that 24-h persistent H_2_O_2_ exposure significantly reduced viability of ARPE-19 cells at doses higher than 125 μM and induced dose-dependent cytotoxicity (Figure 1A). Meanwhile, both cytosolic and released levels of miR-100 in response to H_2_O_2_ treatment were profiled by reverse transcription and quantitative real-time PCR (RT-qPCR) detection. The results indicated that both cytosolic and released contents of miR-100 significantly increased in a dose-dependent manner by oxidative stress (Figure 1B,C). This evidence substantially links the miR-100 upregulation with oxidative cell death of ARPE-19 cells.

### 2.2. Signaling Profiles of Oxidative Stress-Driven Cytotoxicity of ARPE-19 Cells

Since a broad spectrum of signal pathways including PI3K/Akt, MAPK, and NF-κB are involved in the pro-inflammatory and pro-survival responses in RPE cells against oxidative stress [37,38,39,40,41], we next scrutinized the signaling profiles in H_2_O_2_-treated ARPE-19 cells by measuring total and phosphorylated levels of signaling mediators (Figure 2A). Western blotting and densitometrical analysis indicated that H_2_O_2_ treatment prominently elicited instant hyperphosphorylation of Akt, ERK1/2, JNK, and p38 MAPK (Figure 2B–E), peaking within 1 h after the addition of H_2_O_2_. By contrast, H_2_O_2_ induced NF-κB hyperphosphorylation occurred at around 3 to 6 h after addition (Figure 2F). These data confirmed the responsiveness of these signaling pathways to the oxidative-stress-induced toxicity of ARPE-19 cells.

### 2.3. Signal Pathways Involved in Oxidative-Stress-Induced miR-100 Biosynthesis and Cytotoxicity in ARPE-19 Cells

To determine which signal pathway is involved in the oxidative-stress-induced miR-100 biosynthesis in RPE cells, ARPE-19 cells were pretreated with different selective kinase inhibitors and exposed to H_2_O_2_ addition, followed by RT-qPCR detection for miR-100 levels in lysates and condition media. Detection results clearly showed that inhibition of PI3K and p38 MAPK activities partially attenuated H_2_O_2_-upregulated cytosolic miR-100 contents, while NF-κB signal blockade completely suppressed this upregulation (Figure 3A). Similar trends were noted in miR-100 levels of conditioned media in that PI3K/Akt and p38 inhibition abolished H_2_O_2_-increased miR-100 release from ARPE-19 cells (Figure 3B). Intriguingly, NF-κB signal blockade did not affect the increased release of miR-100, whereas JNK inhibition conversely potentiated the H_2_O_2_-induced miR-100 release.

To further examine the involvement of signaling pathways in the oxidative-stress-induced RPE cytotoxicity, APRE-19 cells were again pretreated with kinase inhibitors and received H_2_O_2_ exposure, followed by cytotoxicity assay. Cell viability data indicated that the abolishment of Akt, JNK, and NF-κB signaling activation did not affect persistent H_2_O_2_ exposure-induced cytotoxicity (data not shown), but significantly ameliorated transient H_2_O_2_ treatment-induced cytotoxicity in ARPE-19 cells (Figure 4). Taken with the above miR-100 changes in the treated cells, these findings suggest that both PI3K/Akt and NF-κB signal axes are involved in the oxidative stress-elicited miR-100 changes and cell death of RPE cells.

### 2.4. Suppressive Effect of miR-100 Antagomir on Oxidative Cell Death in ARPE-19 Cells

Given the positive role of miR-100 in oxidative cell death, we next determined whether miR-100 antagonism protects RPE cells against oxidative stress injury. To this end, ARPE-19 cells were transfected with miR-100 antagomir or non-specific miRNA oligonucleotides for 24 h, followed by another 24-h H_2_O_2_ exposure. RT-qPCR measurement confirmed that H_2_O_2_ exposure markedly increased miR-100 levels in both cytosol and conditioned media, while miR-100 levels were not detectable in both compartments after miR-100 antagonistic transfection (Figure 5A). Meanwhile, cytotoxicity assay data demonstrated that miR-100 antagonism significantly reduced the H_2_O_2_-induced cell death in ARPE-19 cells (Figure 5B). In addition, the result of a terminal deoxynucleotidyl transferase dUTP nick end labeling (TUNEL) assay supported that miR-100 antagonism effectively alleviated oxidation-related apoptotic death of ARPE-19 cells (Figure 5C,D). These findings strongly suggest that miR-100 is a novel target for preventing oxidative RPE cell death.

### 2.5. Effects of miR-100 Antagomir on AMPK/mTOR and Antioxidant Expression in ARPE-19 Cells

Given that the AMPK-dependent upregulation of heme oxygenase-1 (HO-1) antioxidant expression protects ARPE-19 cells against oxidative-stress-induced apoptosis [42], we next examined whether AMPK/mTOR axis and HO-1 expression are involved in the miR-100 antagomir exhibited cytoprotection against oxidative injury of RPE cells. Protein lysates collected from the cells receiving an miR-100 antagonist or control miRNA and subsequent H_2_O_2_ exposure were subjected to western blotting detection (Figure 6A). The blotting and densitometry data clearly showed that H_2_O_2_ exposure prominently induced AMPK hyperphosphorylation, while miR-100 antagonistic treatment completely abolished this induction (Figure 6B). By contrast, H_2_O_2_ treatment reduced total protein levels of mTOR, while miR-100 antagomir intriguingly reversed this reduction (Figure 6C), despite no apparent effect on mTOR phosphorylation levels (data not shown). Moreover, the miR-100 antagomir significantly elevated constitutive HO-1 expression and augmented the H_2_O_2_-induced HO-1 upregulation in ARPE-19 cells (Figure 6D), strongly suggesting activation of the Nrf-2 signal in the cells receiving the antagomir treatment under oxidative stress condition. Given that p62 sequestosome 1 (p62/SQSTM1) is reported to protect RPE cells against the oxidative stress associated with cell death via activating the Nrf2/HO-1 axis [43,44], the miR-100 antagomir treatment dramatically restored cellular p62 expression in the cells under oxidative stress (Figure 6E). Immunofluorescent staining also demonstrated that the increased p62 peptides were mainly distributed in perinuclear subcellular locations in the treated cells (Figure 7). Taken together, these findings suggested that not only restoration of cellularAMPK/mTOR levels but also activation of p62/Nrf-2/HO-1 signal axis are involved in the cytoprotective effect of miR-100 inhibition against the oxidative injury in RPE cells.

## 3. Discussion

The pro-apoptotic role of miR-100 has been previously demonstrated in many diseases, in particular extensively discussed in the regulation of tumorigenesis. However, the expression and function of miR-100 vary among different cell types in the tumors. Earlier clinical observatory studies report that miR-100 expression is downregulated in cancer cells, while this miR-100 downregulation enhances tumorigenesis. In the mechanistic context, miR-100 functionally suppresses proliferation and motility [25], induces apoptosis, and enhances the chemosensitivity of cancer cells [26,27]. Conversely, an in vivo miRNA study identifies that loss of miR-100 function in tumor cells enhances clonogenic survival and results in the reduction of apoptosis and radiosensitivity [28]. By contrast, it is reported that abundant expression of miR-100 in exosomes secreted from prostate cancer stimulates matrix metalloproteinase expression, thereby contributing to the pro-metastatic niche during tumor development [45]. Moreover, the highly expressed miR-100 maintains the phenotype of tumor-associated macrophages by targeting mTOR to promote tumor metastasis in mice, while intratumoral injection with an miR-100 inhibitor and combined cisplatin therapy showed better benefits for breast cancer therapy [29]. These findings strongly suggest that, under pathogenetic and oxidative conditions, the vitreous and endogenous miR-100 levels may play an essential role in determining the cell fate of resident retinal cells including the RPE cells. In this regard, more clinical and preclinical studies may further warrant the pathogenetic role of miR-100 in AMD.

Recent advances have emphasized the importance of oxidative stress in the pathogenesis of AMD, thus numerous novel therapeutics were designed to alleviate apoptotic cell death of RPE by supplementing antioxidant or raising antioxidant expression in retinal tissues. Despite the fact that miRNA dysregulation is a common event in human diseases including cancers and ocular diseases, the present study demonstrated that biosynthesis and release of miR-100 increased in cultured RPE cells under oxidative stress, thereby regulating cellular apoptosis. Our findings not only support the involvement of miR-100 elevation in the pathogenesis of AMD and unravel the signaling mechanisms governing miR-100 biosynthesis, but also clarify the anti-apoptotic effect of miR-100 antagomir treatment in cultured RPE cells. Collectively, miR-100 is strongly suggested as a potential therapeutic target in the prevention of oxidative RPE cell death and AMD treatment.

The findings of this study delineate the signal network elicited by H_2_O_2_-reduced oxidative stress and its possible connection to endogenous miR-100 biosynthesis and consequent apoptotic cell death of RPE cells (Figure 8). The findings of kinase inhibition experiments determining how the miR-100 expression and release is regulated within the cells suggest the involvement of PI3K/Akt, p38 MAPK, and NF-κB pathways in its biosynthesis, while Akt and p38 signal activities participate in the regulation of its release from the cells. Due to the concern for the potential non-specificity of kinase inhibitors, further study by using genetic intervention tools may warrant the validity of this result. Since p38 MAPK reportedly mediates adenosine-induced alterations in myocardial glucose utilization via AMPK, which is thus claimed to likely lie downstream of p38 MAPK, p38 inhibition restores adenosine-induced cardioprotection in the stressed hearts [46]. However, the present study demonstrated that p38 inhibition alleviated H_2_O_2_-increased miR-100 biosynthesis in, and its release from, ARPE-19 cells but contradictorily did not protect them from oxidative cell death. In contrast, inhibition of Akt, JNK, and NF-κB signaling activities significantly reversed the H_2_O_2_-reduced cell viability, implicating that miR-100 biosynthesis is not the sole player in cell fate determination. Among these pathways, blockade of the NF-κB signal axis might be the most potent player that suppresses the oxidative-stress-induced miR-100 biosynthesis and subsequent RPE cell death. Moreover, the delayed phosphorylation in response to H_2_O_2_ exposure implicates that the NF-κB signal mediator likely lies downstream of the Akt and JNK axes and converges both signaling activities, so that its signal blockade elicits a better cytoprotective effect. However, the discrepancy between the miR-100 biosynthesis and cell viability upon JNK inhibition under oxidative stress has not been resolved to date [47] and more studies may warrant further elucidation.

In the mechanistic context of oxidative retinal injury, H_2_O_2_ is widely known to activate AMPK and inhibit mTOR activation in ARPE-19 cells [38,39,40]. Given that AMPK is an upstream negative regulator of mTOR [48], the PI3K/Akt and mTOR axes are known to mediate RPE cell survival during oxidative-stress-induced apoptosis [37]. In this regard, the α-melanocyte-stimulating hormone is reported to trigger Akt/mTOR pro-survival signaling in the RPE cells under oxidative stress [39], while the rapamycin-sensitive mTOR activation is found to mediate nerve growth factor-induced cell migration and pro-survival signals against oxidative stress in RPE cells [40]. In fact, the tumor-suppressive effect of miR-100 has long been recognized to lie in its targeting effect on mTOR expression in tumor cells, thereby modulating chemo- and radio-sensitivities [27]. miR-100 additionally inhibits IGF-1R expression not only in hepatocellular carcinoma cells [30] but also in retinal ganglion cells [33], highlighting the significance of miR-100 in regulating the IGF-1R/Akt-mediated pro-survival signals and implicating its clinical applicability in preventing retinal ganglion cell death. 

More importantly, this study uncovers the relationship between miR-100 and HO-1 antioxidant expression in RPE cells. Consistent with the protective role of mTOR complex 1 signaling activation and subsequent Nrf2/HO-1 axis against oxidative-stress-induced RPE cell death [42], this study presents the first evidence that miR-100 antagomir treatment concurrently attenuates oxidative stress-related AMPK phosphorylation, restores cellular mTOR and p62 levels, and upregulates HO-1 expression in ARPE-19 cells. The elevation of p62 expression under oxidative stress is found to disrupt inhibitory interaction between Nrf2 and Keap1 proteins, thereby leading to increased expression of Nrf2 target genes and consequent antioxidant response [43,44]. In addition to the regulatory role of the Nrf2/HO-1 axis in RPE cells, the p62 sequestosome protein also recognizes ubiquitinated perinuclear aggregates and the p62-tagged materials can be eradicated from cytosol [49]. Therefore, the dual cytoprotective mechanisms of p62 restoration through the activation of Nrf2 signaling and autophagy clearance of misfolded proteins have been claimed to be a potential target for preventing oxidative cell death of RPE and treating AMD [43]. While the role of p62 expression has been linked with autophagy function, scavenger receptor expression, and phagocytosis [50,51], the effect of miR-100 antagomir treatment on autophagic and phagocytic functions of RPE deserves further elucidation. Moreover, a negative correlation between miR-100 and p62 expression is significantly noted in hepatocellular carcinoma cells [30], supporting their reciprocal regulation in the cells. Taken together, the restoration of cellularAMPK/mTOR levels and the activation of p62/Nrf-2/HO-1 signal axis miR-100 antagonistic agents strongly support the therapeutic applicability in the AMD. 

The pro-angiogenic role of miR-100 previously reported in tumor formation may provide mechanistic insights on the regulatory relationship between miR-100 and signaling activity in the RPE cells. In this context, miR-100 enrichment in mesenchymal stem cell-derived exosomes has been found to mediate VEGF expression in breast cancer cells [52]. In the context of ocular angiogenesis, miR-100 is constitutively and highly expressed in retinal microvascular endothelial cells, but its expression is suppressed by angiogenic stimulation [34], suggesting differential roles of miR-100 in promoting tumoral angiogenesis but suppressing retinal neovascularization. Nevertheless, the involvement of miR-100 in repressing mTOR signaling has been previously identified in endothelial and vascular smooth muscle cells [53], consistently supporting its anti-angiogenic role as well as its suppressive effect on mTOR signaling in ARPE-19 cells. Moreover, an animal study by using streptozotocin-induced diabetic retinopathy corroborates that upregulation of NF-κB- and VEGF-responsive miRNAs in the retina and retinal endothelial cells constitute a key miRNA signature that contributes to pathologic changes of early diabetic retinopathy [54], implicating a negative feedback regulation of NF-κB activation during ocular neovascularization. Furthermore, our findings clearly showed that NF-κB signaling inhibition completely suppressed H_2_O_2_-induced miR-100 biosynthesis in ARPE-19 cells, suggesting that miR-100 very likely plays distinct roles in the pathogeneses of dry and wet AMD. However, clinical comparative study on miR-100 profiles in both types of AMD may warrant further elucidation. In conclusion, this study demonstrated that miR-100 mediates oxidative cell death of ARPE-19 cells and the downregulation of miR-100 via antagonistic agents may constitute a therapeutic strategy for the treatment of AMD.

## 4. Materials and Methods

### 4.1. Reagents

Selective kinase-specific inhibitors including LY294002 for PI3K, SB203580 for p38 MAPK, PD98059 for MAPK/ERK kinase 1 (MEK1), SP600125 for JNK, and PDTC for NF-κB inhibition were purchased from Sigma-Aldrich (St. Louis, MO, USA). All inhibitors were dissolved in DMSO and stocked at 10 mM at −20 °C. Antibodies raised against phospho-Akt (Ser 473), phospho-ERK1/2 (Thr202/185 and Tyr204/187), phospho-p38 MAPK (Thr180/Tyr182), phospho-JNK (Thr183/Tyr185), phospho-NF-κB (Ser536), phospho-AMPKα1 (Thr172), phospho-mTOR (Ser2448), and those against respective total proteins were purchased from Cell Signaling (Beverly, MA, USA). Antibody against HO-1 was purchased from Enzo Life Sciences Inc. (Farmingdale, NY, USA) and that against β-Actin was from Merck Millipore (Temecula, CA, USA).

### 4.2. Cell Culture and Treatment

A human RPE cell line, ARPE-19, was obtained from American Type Culture Collection (Manassas, VA, USA) and cultured in a Dulbecco’s modified Eagle’s medium (DMEM) and Ham’s F12 1:1 nutrient mixture medium supplemented with 10% fetal bovine serum and regular antibiotics (Invitrogen/Gibco, Gaithersburg, MD, USA) in a humidified atmosphere containing 5% CO_2_ at 37 °C. The cells were previously authenticated by short tandem repeat DNA profiling analysis (Appendix A) and used for study within three passages after thawing. To mimic oxidative stress, ARPE-19 cells were treated with H_2_O_2_ dissolved in a medium at the indicated dose for either persistent 24-h or transient 2-h periods, the cellular lysates and supernatants were collected and subjected to detection assays as described below.

### 4.3. Cytotoxicity Assay

A 3-(4, 5-dimethylthiozol-2-yl)-2, 5-diphenyltetrazolium bromide (MTT) (Sigma- Aldrich, St. Louis, MO, USA) based cell proliferation assay was used to measure the cytotoxic effect of oxidative on ARPE-19 cells as previously described [55]. In brief, ARPE-19 cells were plated in 96 well plates at a density of 2 × 10^5^ cells/cm^2^. At the end of MTT (500 μg/mL) incubation, optical density was measured at 540 nm by a microplate reader (Sunrise™, Tecan, Mannedorf, Switzerland). All groups were tested in triplicate and the cells without chemical treatment were used as negative controls and the wells without cells as blank controls.

### 4.4. MiRNA Semi-Quantification

miRNA was extracted from cell lysates and conditioned media by using a High Pure miRNA Isolation Kit (Roche Applied Science, Mannheim, Germany) according to the manufacturer’s instructions. The collected supernatants were centrifuged to remove floating cells and debris prior to detection. Subsequent RT-qPCR was performed by a TaqMan Advanced miRNA Assay Kit (Thermo Fisher Scientific Inc., Waltham, MA, USA) using an Eco Real-Time PCR System (Illumina, San Diego, CA, USA) according to the manufacturer’s instructions. Cellular miR-361 level in H_2_O_2_-treated cells was verified stable and used as internal controls. Expression levels of miR-100 were normalized to the endogenous miR-361 level and presented as folder changes (2-^ΔΔCt^) against the negative control. A medium-alone control was used to exclude the pre-existence of miR-100 in the culture medium.

### 4.5. Western Blotting

After drug treatment, ARPE-19 cells were rinsed with ice-cold PBS and lyzed in prechilled RIPA buffer with the addition of protease and phosphatase inhibitors. After protein quantification, 50 µg of total cell lysates were electrophoresed, electrotransferred, and immunodetected using the previously described protocol [56]. The immunoreactive signals were documented with the enhanced chemiluminescence detection system (Millipore, Temecula, CA, USA). Band density was measured using ImageJ image analytical software (National Institutes of Health, Bethesda, MD, USA). The relative expression levels of target proteins are shown by the density ratio to β-Actin internal control or to unphosphorylated total protein in the same sample. All density results were obtained from three independent experiments and normalized as percentage or induction folds of negative control level.

### 4.6. TUNEL Staining

ARPE-19 cells were fixed with ice-cold paraformaldehyde after receiving treatment and subjected to a TUNEL-based in situ cell death detection assay. Fluorescent TUNEL assay was performed according to the standard protocol provided by the manufacturer (Roche, Mannheim, Germany). After staining, a mounting medium containing DAPI was used to identify nuclei of cells. Quantification of fluorescent positive signals in each group was performed by counting at least 10 randomly selected images at high-power fields under fluorescent microscopy and the TUNEL positivity was shown in the percentage of total cells.

### 4.7. Transfection of miR-100 Inhibitor

Single-stranded and RNA-based oligonucleotides specifically targeting hsa-miR-100-5p (miR-100 inhibitor) were synthesized by Ambion (Thermo Fisher Scientific Inc., Waltham, MA, USA). ARPE-19 cells were transfected with a 200 nM miR-100 inhibitor by using Lipofectamine 2000 24 h before H_2_O_2_ treatment. In control experiments, cells were transfected with 200 nM non-specific control miRNA.

### 4.8. Immunofluorescent Staining

ARPE-19 cells, after treatments, were fixed with ice-cold methanol and permeabilized with PBS containing 0.2% Triton X-100. The cells were then blocked with 1% BSA and stained with 2 μg/mL anti-p62/SQSTM1 antibody (Santa Cruz Biotechnology, CA, USA) for overnight incubation at 4 °C, followed by treatment with Alexa488-conjugated anti-mouse IgG for 1 h at room temperature in the dark. After washes, the cells were counterstained with 1 μg/mL DAPI for 10 min, mounted with DAKO fluorescent mounting fluid, and observed under a fluorescent microscope (Axiovert, Zeiss, Germany).

### 4.9. Statistical Analysis

All results are expressed as the mean ± SD. Data were analyzed by using an unpaired Student’s t-test with two-tailed analysis or one-way ANOVA followed by Tukey’s post hoc test. *p*-values < 0.05 are considered statistically significant.

## 5. Conclusions

In conclusion, this study demonstrated that the H_2_O_2_ -mimicked oxidative stress in cultured ARPE-19 cells may trigger miR-100 biosynthesis that mediates apoptotic cell death, while the signal pathways including PI3K/Akt, p38 MAPK, and NF-κB were involved in oxidative-stress-induced miR-100 upregulation in the cells. Conversely, miR-100 antagomir treatment exerted a cytoprotective effect against the H_2_O_2_-induced oxidative cell death through attenuating the oxidation-induced AMPK hyperphosphorylation, restoring cellular mTOR and p62 levels and upregulating heme oxygenase-1 expression. Our findings strongly support that miR-100 antagonism can be regarded as a preventive/therapeutic strategy for treatment of retinal degenerative disease.

## Figures and Tables

**Figure 1 pharmaceuticals-14-00314-f001:**
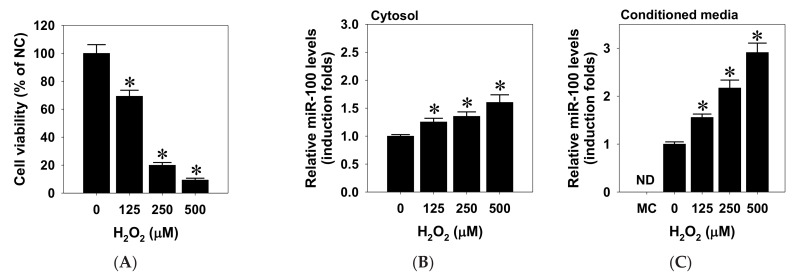
Hydrogen peroxide treatment induces miR-100 biosynthesis in human ARPE-19 cells. (**A**) ARPE-19 cells were treated with H_2_O_2_ at indicated doses for 24 h and subjected to cellular viability assay. (**B**,**C**) Alternatively, biosynthesis and release of miRNA was measured by isolating miRNA from lysates and conditioned media, followed by miRNA-100 quantification using RT-qPCR. All data are shown in mean ± SD from three independent experiments. * *p* < 0.05 vs. zero control by *t*-Test. MC, medium control; ND, not detectable.

**Figure 2 pharmaceuticals-14-00314-f002:**
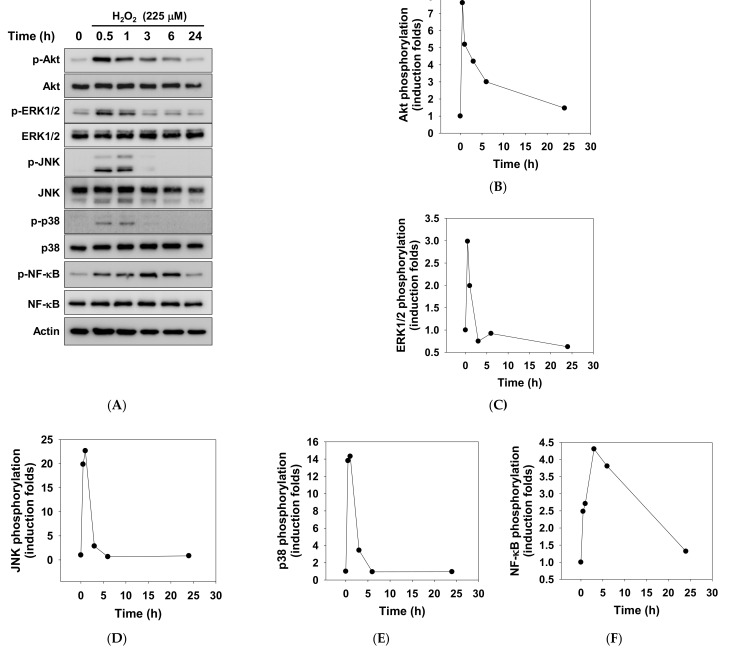
Characterization of hydrogen peroxide-driven signaling kinetics in human ARPE-19 cells. (**A**) ARPE-19 cells were treated with 225 μM of H_2_O_2_ for the indicated time. Cellular protein lysates were subjected to western blotting detection for total and phosphorylated kinase proteins. Densitometric analysis was used to quantify phosphorylation levels of Akt (**B**), ERK1/2 (**C**), JNK (**D**), p38 MAPK (**E**), and NF-κB (**F**). Data are expressed as mean values from three independent experiments.

**Figure 3 pharmaceuticals-14-00314-f003:**
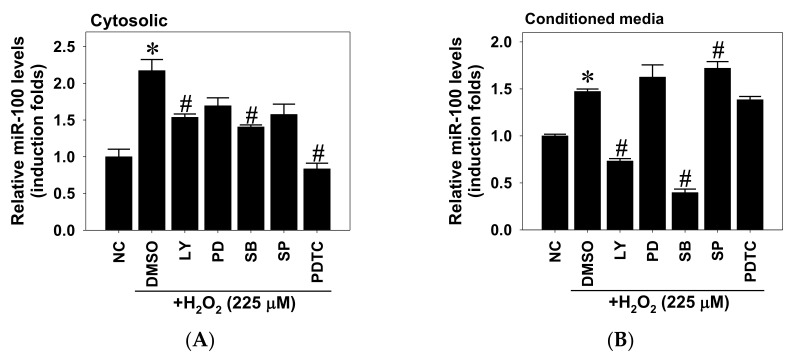
Involvement of hydrogen peroxide-activated signaling in induced miR-100 biosynthesis in human ARPE-19 cells. ARPE-19 cells were pretreated with kinase-specific inhibitors for 1 h, including the PI3K inhibitor LY294002 (LY, 10 µM), MEK1 inhibitor PD98059 (PD, 10 µM), p38 inhibitor SB203580 (SB, 20 µM), JNK inhibitor SP600125 (SP, 20 µM), NF-κB inhibitor pyrrolidine dithiocarbamate (PDTC, 50 µM), or equivalent dimethyl sulfoxide (DMSO) as solvent control, followed by H_2_O_2_ exposure at 225 µM. After 24 h of treatment, lysates (**A**) and conditioned media (**B**) were subjected to miRNA isolation and subsequent RT-qPCR detection for miRNA-100 levels. Semi-quantitative mRNA levels are shown as an induction fold of negative control (NC) in mean ± SD from three independent experiments. * *p* < 0.05 vs. NC group; # *p* < 0.05 vs. DMSO groups by ANOVA.

**Figure 4 pharmaceuticals-14-00314-f004:**
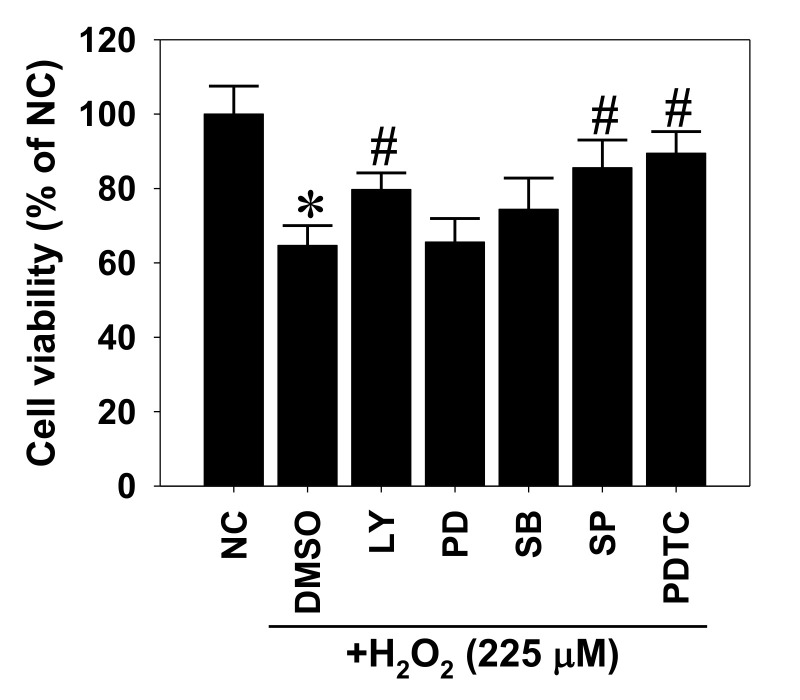
Involvement of hydrogen peroxide-activated signaling in its cytotoxic effect on human ARPE-19 cells. ARPE-19 cells were pretreated with kinase-specific inhibitors for 1 h, including the PI3K inhibitor LY294002 (LY, 10 µM), MEK1 inhibitor PD98059 (PD, 10 µM), p38 inhibitor SB203580 (SB, 20 µM), JNK inhibitor SP600125 (SP, 20 µM), NF-κB inhibitor pyrrolidine dithiocarbamate (PDTC, 50 µM), or equivalent dimethyl sulfoxide (DMSO) as solvent control, followed by transient H_2_O_2_ exposure at 225 µM for 2 h. After another 22-h incubation, the treated cells were subjected to cytotoxicity assay. Cell viability is expressed as a percentage of negative control (NC) in mean ± SD from three independent experiments. * *p* < 0.05 vs. NC group; # *p* < 0.05 vs. DMSO groups by ANOVA.

**Figure 5 pharmaceuticals-14-00314-f005:**
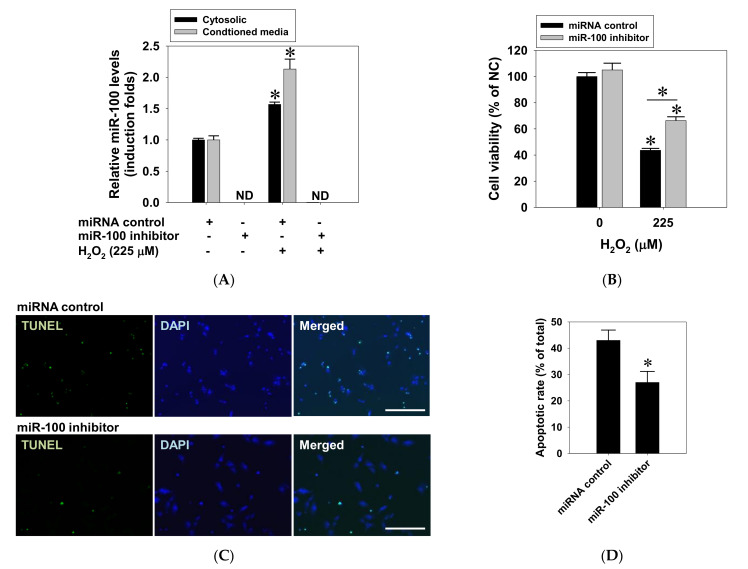
Mitigation of hydrogen peroxide-induced apoptosis of human ARPE-19 cells by miR-100 antagonism. ARPE-19 cells were transfected with either a 200 nM miR-100 antagonist or control oligonucleotides for 24 h followed by H_2_O_2_ exposure at 225 µM for another 24 h. (**A**) Cellular lysates and conditioned media were collected for miRNA isolation and miR-100 detection by RT-qPCR. ND, not detectable. (**B**) Cytotoxicity assay was used to determine the viability of the treated cells. Alternatively, the treated cells were fixed and subjected to fluorescent TUNEL staining (**C**) and subsequent analysis on TUNEL staining positivity (**D**). Data are expressed as mean ± SD from three independent experiments. * *p* < 0.05 vs. control or between indicated groups, by ANOVA. Scale bar = 50 µm. +/− indicates presence/absence of reagents.

**Figure 6 pharmaceuticals-14-00314-f006:**
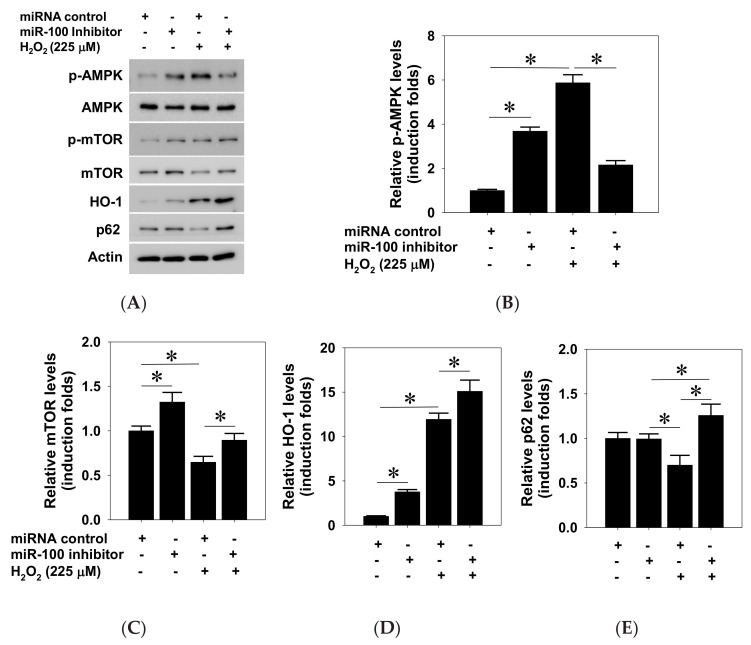
Effects of miR-100 inhibition on AMPK/mTOR signaling and heme oxygenase 1 (HO-1) expression in human ARPE-19 cells. ARPE-19 cells were transfected with either an miR-100 inhibitor or negative miRNA control nucleotides for 24 h followed by H_2_O_2_ exposure at 225 μM for another 24 h. After treatments, cellular protein lysates were collected for western blotting detection (**A**) and subsequent densitometry analysis for p-AMPK (**B**), total mTOR (**C**), HO-1 (**D**), and p62 levels (**E**). Data are expressed as mean ± SD from three independent experiments. * *p* < 0.05 between indicated groups by ANOVA. +/− indicates presence/absence of reagents.

**Figure 7 pharmaceuticals-14-00314-f007:**
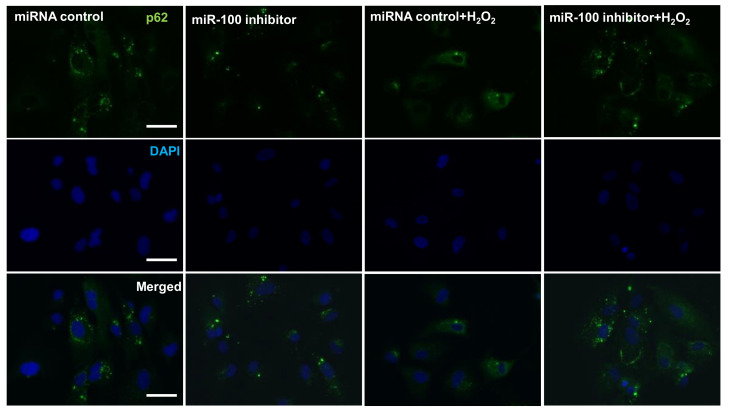
Effects of miR-100 inhibition on cellular expression and distribution of p62 in human ARPE-19 cells under oxidative stress. ARPE-19 cells were transfected with either miR-100 inhibitor or miRNA control nucleotides for 24 h followed by H_2_O_2_ exposure at 225 μM for another 24 h. Afterward, the cells were fixed and subjected to immunofluorescent staining. Note that miR-100 inhibition restored autophagy-associated perinuclear granules in the H_2_O_2_-exposed cells. Scale bar = 20 μm.

**Figure 8 pharmaceuticals-14-00314-f008:**
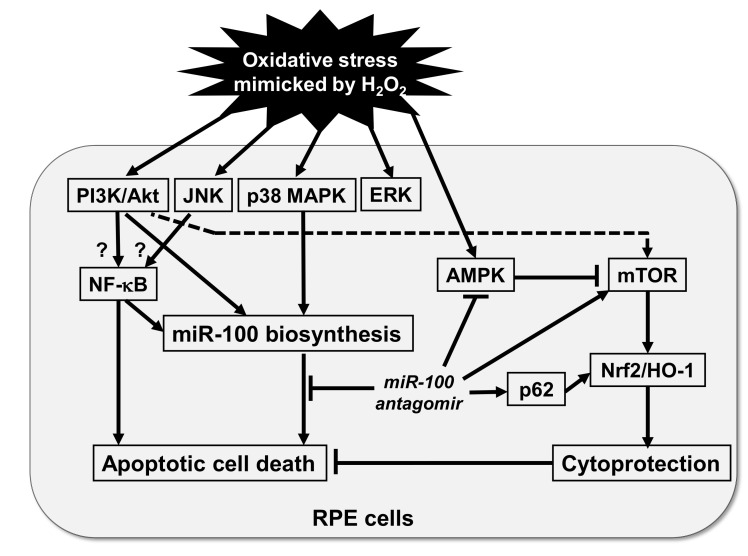
A hypothetical scheme showing the network of oxidative-stress-induced signal pathways, miR-100 biosynthesis, and their roles in the oxidative cell death of retinal pigment epithelial (RPE) cells. The study findings demonstrated that hydrogen peroxide (H_2_O_2_)-mimicked oxidative stress may activate signaling pathways including the hyperphosphorylation of Akt, JNK, p38 MAPK, ERK, and NF-κB. The mediator activities of PI3K/Akt, p38 MAPK, and NF-κB contribute to endogenous miR-100 expression, thereby leading to apoptotic cell death. The NF-κB signal mediator very likely lies downstream of the Akt and JNK axes, which awaits further elucidation. The PI3K/Akt/mTOR axis is known to mediate cell survival signals. Mechanistically, miR-100 antagomir treatment not only attenuates oxidative stress-related AMPK phosphorylation but also restores cellular mTOR and p62 levels and upregulates antioxidant heme oxygenase-1 (HO-1) expression, thereby protecting RPE cells against oxidative cell death. Arrows, stimulatory; T lines, inhibitory; Solid lines, new findings of this study; Dashed line, previously identified mechanism; Question marks, unknown mechanisms.

## Data Availability

The data presented in this study are available on request from the corresponding author.

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
