# Peer review of "MicroRNA-100 Mediates Hydrogen Peroxide-Induced Apoptosis of Human Retinal Pigment Epithelium ARPE-19 Cells"

_pharmaceuticals, 2021, doi:10.3390/ph14040314_

Round 1

Reviewer 1 Report

Chang et al., attempt to describe the role of MicroRNA-100 in human RPE apoptosis under oxidative stress as related to AMD. The study is based on a previous study that described a similar role in RGC’s apoptosis. Such studies implicating miR-100 in cancers is also well established. The authors used ARPE-19 cells to hypothesize that miR1-00 plays a regulatory role in oxidative stress-induced apoptosis in RPE. After selecting the dose and time of H2O2 that cause decreased cell viability, the authors tested the upregulation of mIR-100. Subsequently, the authors tested miR-100 antagonist to establish various cell signaling pathways downstream of miR-100. Particularly authors showed AMPK/mTOR/HO-1 signaling altered with miR-100 inhibition in the presence of H2O2. Authors reached their conclusion that “miR-100 mediates oxidative cell death of ARPE-19 cells and downregulation of miR-100 via antagonistic agents may constitute a therapeutic strategy for treatment of AMD”.

  1. For the most part, this is a well-written manuscript with impressive figures. However, the manuscript lacks novelty, and if any, the data shown is incremental in nature. Perhaps authors should consider some functional studies including but not limited to phagocytosis.
  2. As authors have already acknowledged, the role of miR-100 in apoptosis is well documented both in cancers and in particular ocular diseases. It is unclear why RPE should be the target of miR-100, specifically in AMD. In the absence of any pre-clinical or clinical evidence that RPE is the target of miR-100, the data extrapolated from RGC’s or other cancer cell types is not justified.
  3. The pharmacological inhibitors used in the study are well documented to be non-specific. A more straightforward study could involve the use of dominant-negative constructs to tease out specific signaling downstream of miR-100. As presented, the data is diffuse.
  4. In the absence of clear evidence that conditioned medium has extracellular vesicles that might carry micro RNA, the data shown from Fig 1C is not convincing. It is also not clear how authors have distinguished cytosolic and conditioned medium fractions and how the miR-100 expression is regulated within the cell to be secreted.
  5. Figure 5, the quantification of apoptosis may be subjective with a method involving microscopic images. A more robust quantification is necessary.
  6. The authors must identify which of the data used t-tests and which one’s ANOVA.
  7. Since expression levels of miR-100 were normalized to endogenous miR-361 level, authors must validate that the endogenous control is unchanged with H2O2 treatment in RPE cells.

Author Response

Point 1: For the most part, this is a well-written manuscript with impressive figures. However, the manuscript lacks novelty, and if any, the data shown is incremental in nature. Perhaps authors should consider some functional studies including but not limited to phagocytosis.

Response 1: We thank for reviewer's suggestion to test additional functional experiment such as phagocytosis. However, due to the time limitation, only 10 days for revision requested by editor, we have to apologize for our inability to perform this experiment.

Point 2: As authors have already acknowledged, the role of miR-100 in apoptosis is well documented both in cancers and in particular ocular diseases. It is unclear why RPE should be the target of miR-100, specifically in AMD. In the absence of any pre-clinical or clinical evidence that RPE is the target of miR-100, the data extrapolated from RGC’s or other cancer cell types is not justified.

Response 2: We agree reviewer with his/her viewpoint on the study background. We have revised texts to reveal the situation by adding some statements in Introduction (line 75) and Discussion sections (line 243).

Point 3: The pharmacological inhibitors used in the study are well documented to be non-specific. A more straightforward study could involve the use of dominant-negative constructs to tease out specific signaling downstream of miR-100. As presented, the data is diffuse.

Response 3: We agree with reviewer's opinion in the non-specificity issue of some kinase inhibitors and thank for his/her precious suggestion. However, we again apologize for our inability to perform dominant-negative construct experiment due to the limitation of revision time.

Point 4: In the absence of clear evidence that conditioned medium has extracellular vesicles that might carry micro RNA, the data shown from Fig 1C is not convincing. It is also not clear how authors have distinguished cytosolic and conditioned medium fractions and how the miR-100 expression is regulated within the cell to be secreted.

Response 4:

(1) To answer reviewer's first question, we found two references supporting extracellular vesicles carrying miRNA and accordingly revised the Results text (section 2.1, line 87) by adding a statement, "Recent studies have evidenced that the existence of miRNA in the extracellular vesicles released by RPE cells may contribute to retinal pathophysiolgical processes [34,35]."

(2) For the question regarding to the technique for collecting cytosolic and conditioned medium fractions, it is undoubted that a simple high speed centrifugation could easily remove the suspension cells and debris from the conditioned medium.

(3) As for how the expressed miR-100 is regulated for its extracellular release, this study did not indeed provide detailed mechanism. We agree with reviewer that it is an interesting issue to be solved further, and also believe it won't affect our conclusion.

Point 5: Figure 5, the quantification of apoptosis may be subjective with a method involving microscopic images. A more robust quantification is necessary.

Response 5: We thank for reviewer's suggestion and agree that a more robust quantification for the apoptotic signal could be done, such as quantifying the fluorescent intensity of apoptotic cells. Again, we apologize for our inability to confirm this data due to the time limitation.

Point 6: The authors must identify which of the data used t-tests and which one’s ANOVA.

Response 6: We thank reviewer for addressing this statistical analysis issue. We have added descriptions to specify the statistical methods in figure legends.

Point 7: Since expression levels of miR-100 were normalized to endogenous miR-361 level, authors must validate that the endogenous control is unchanged with H2O2 treatment in RPE cells.

Response 7: We thank reviewer for pointing out the miRNA quantitative issue. We have preliminarily checked the endogenous miR-361 levels in all groups and confirmed relatively stable Ct values in each treatment group (value changes less than 0.5). We accordingly added statement in Methodology section 4.4 to describe the stable expression of miR-361 levels as internal control for all experimental treatments.

Reviewer 2 Report

Chang and coworkers analyzed the role and mechanism of miRNA-100 on hydrogen peroxide-induced apoptosis of ARPE-19 cells. To this end, they could show an enhanced expression of miRNA-100 following treatment of ARPE-19 cells with hydrogen peroxide. Further on, they observed an activation of AKT, ERK1/2, JNK, p38 and NF-(kappa)B after treatment of the cells. The inhibition of the AKT, p38 and NF-(kappa)B pathways leads to an inhibited miRNA-100 induction and an increased survival of ARPE-19 cells following hydrogen peroxide treatment. Intriguingly, the inhibition of miRNA-100 expression in ARPE-19 cells promotes cell survival and blocks apoptosis after treatment. Finally, the authors describe that the inhibition of miRNA-100 leads to an enhanced expression of HO-1, which is known to be antioxidative.

Overall, the manuscript is of high interest and the authors analyzed the role of miRNA-100 in stressed ARPE-19 cells in an appropriate manner. However, some points should be considered before publication.

1) Even the written English of the manuscript is not bad, there are some spelling mistakes, duplications and odd expressions. I would recommend a proof reading by a native speaking colleague.

2) If native RPE cells are available, I would suggest to repeat the milestone experiments, such as induction of miRNA-100 following hydrogen peroxide treatment, with those cells too.

3) in paragraph 2.2 the authors conclude “These data confirmed the responsibility of these signaling to oxidative stress injury in ARPE-19 cells.” This conclusion is not correct, since these data only show that the signaling pathways are activated by oxidative stress. Please correct.

4) In figure 3 and 4 the authors only show the inhibitors of the pathways and the explanation, which inhibitor blocks the respective pathway, is only given in paragraph materials and methods. It would be more comfortable for the reader, if the authors explain the inhibitors in the figures legend.

5) To analyze the role of the investigated signaling pathways on cell survival (figure 4) the authors incubated the cells for 2 h with hydrogen peroxide. In all other experiments the cells were treated for 24 h. Why was the exposure time shortened for this experiment?

6) In the legend of figure 5 the authors describe images from A to E, but in figure 5 only images from A - D are shown. Please correct.

Author Response

Point 1: Even the written English of the manuscript is not bad, there are some spelling mistakes, duplications and odd expressions. I would recommend a proof reading by a native speaking colleague.

Response 1: Thanks for reviewer's suggestion. The manuscript has been edited by native speaker, and spelling mistakes in text have been corrected.

Point 2: If native RPE cells are available, I would suggest to repeat the milestone experiments, such as induction of miRNA-100 following hydrogen peroxide treatment, with those cells too.

Response 2: We thank reviewer for his/her suggestion. However, due to the time limitation, only 10 days for revision requested by editor, we have to apologize for our incapability to perform additional experiments using native RPE cells.

Point 3: in paragraph 2.2 the authors conclude “These data confirmed the responsibility of these signaling to oxidative stress injury in ARPE-19 cells.” This conclusion is not correct, since these data only show that the signaling pathways are activated by oxidative stress. Please correct.

Response 3: We thank reviewer for pointing out this error. We have corrected it "These data confirmed the responsiveness of these signaling pathways to the oxidative stress induced injury in ARPE-19 cells."

Point 4: In figure 3 and 4 the authors only show the inhibitors of the pathways and the explanation, which inhibitor blocks the respective pathway, is only given in paragraph materials and methods. It would be more comfortable for the reader, if the authors explain the inhibitors in the figures legend.

Response 4: We thank reviewer for his/her thoughtful suggestion. We have added description for the kinase inhibitors in the legends of Figures 3 and 4.

Point 5: To analyze the role of the investigated signaling pathways on cell survival (figure 4) the authors incubated the cells for 2 h with hydrogen peroxide. In all other experiments the cells were treated for 24 h. Why was the exposure time shortened for this experiment?

Response 5: We thank reviewer for addressing the different condition for cell survival experiments. Our pilot study showed that kinase inhibition did not improve the cell viability under 24-hour persistent H2O2 exposure (data not shown). That's the reason why we further observed the possible cytoprotective effect under transient 2-hour treatment condition (Figure 4). Accordingly, we have added statement in the Results text section 2.3 (2nd paragraph).

Point 6: In the legend of figure 5 the authors describe images from A to E, but in figure 5 only images from A - D are shown. Please correct.

Response 6: We thank reviewer for pointing out this error. We have corrected the typo.

Reviewer 3 Report

Brief Overview:  Chang YS et al.  investigated the role of microRNA 100 (miR-100) in human retinal pigmented epithelial (ARPE-19) cells treated with hydrogen peroxide (H2O2) to induce oxidative stress.  H2O2 dose-dependently reduced cell viability and this was associated with increased MiR-100.  Transfection of the cells with an miR-100 antagonist preserved cell viability.  The authors conclude that miR-100 is involved in oxidative stress and RPE death.  Therefore, miR-100 antagonists should be considered as therapies for age-related macular degeneration (AMD).

Comments to the Authors:

This work addresses an important health related problem.  The manuscript is well written and the methods employed are appropriate. 

The following clarifications are needed:

  1. State clearly the working hypothesis.
  2. Further details regarding the “transient 2-hour exposure” to H2O2 is needed.  This is mentioned only on page 9, line 314, but no subsequent data provided for these 2-hour exposure experiments.
  3. Further details regarding H2O2 pretreatment is needed.  What was the vehicle used to make the concentrations?  Was the same vehicle used for the “0” group?
  4. Why was H2O2 levels not determined in the cell lysates, particularly after addition of the miR-100 inhibitor? This would have provided key information regarding cell uptake.
  5. Page 3, line 106: H2O2 was not “stimulated”, but was added.  Consider changing to “treatment” or similar.
  6. In the discussion, provide an explanation for the delayed phosphorylation of NFkB compared to the other mediators studied in Figure 2.
  7. In the discussion provide a possible pathway for how MiR-100 is involved in oxidative stress connecting all the mediators studied.
  8. Page 7, line 229: correct the grammatical error “to promotes”.
  9. Are there studies linking specifically H2O2 induction and/or accumulation in the pathogenesis of AMD?  In vivo, H2O2 is rapidly detoxified by glutathione and catalase to oxygen and water.

Author Response

Point 1: State clearly the working hypothesis.

Response 1: As requested by reviewer, we have revised sentences in the Introduction section (on page 2 line 75) to improve the study hypothesis statements.

Point 2: Further details regarding the “transient 2-hour exposure” to H2O2 is needed. This is mentioned only on page 9, line 314, but no subsequent data provided for these 2-hour exposure experiments.

Response 2: We thank reviewer for addressing the different condition for cell survival experiments (this issue is similar to Reviewer#2 Q5). Because our pilot study showed that kinase inhibition did not improve the cell viability under 24-hour persistent H2O2 exposure (data not shown), we further observed the possible cytoprotective effect under transient 2-hour treatment condition (Figure 4). Accordingly, we have added statement in the Results text section 2.3 (2nd paragraph).

Point 3: Further details regarding H2O2 pretreatment is needed. What was the vehicle used to make the concentrations? Was the same vehicle used for the “0” group?

Response 3: We thank reviewer for addressing this technical issue. The vehicle used for H2O2 preparation was basal DMEM, we accordingly omitted the vehicle control group for the experiments. As suggested by reviewer, we have added this information for H2O2 preparation in Methodology section 4.2.

Point 4: Why was H2O2 levels not determined in the cell lysates, particularly after addition of the miR-100 inhibitor? This would have provided key information regarding cell uptake.

Response 4: We thank reviewer for his/her excellent suggestion. As mentioned by reviewer in point 9 below, H2O2 could be rapidly detoxified by glutathione and catalase to oxygen and water. Due to inaccessibility of sensitive instrument for cellular H2O2 detection and the limitation for revision due date, we apologize that we did not perform this experiment. We believe the outcome of miR-100 antagomir treatment in the cell survival still support our conclusion.

Point 5: Page 3, line 106: H2O2 was not “stimulated”, but was added. Consider changing to “treatment” or similar.

Response 5: As suggested by reviewer, we have corrected the sentence to " H2O2 treatment prominently elicited instant hyperphosphorylation of Akt, ..."

Point 6: In the discussion, provide an explanation for the delayed phosphorylation of NFkB compared to the other mediators studied in Figure 2.

Response 6: We thank for reviewer's suggestion. We have added an explanation for the delayed phophorylation of NFkB in the Discussion section (line 274-277).

Point 7: In the discussion provide a possible pathway for how MiR-100 is involved in oxidative stress connecting all the mediators studied.

Response 7: We thank for reviewer's thoughtful opinion. Based on our study findings, we additionally drawn a mechanistic scheme (Figure 7), to delineate the possible network of the oxidative stress induced signaling and miR-100 biosynthesis as well as the connection to pro-survival signal. This figure is also presented as the graphical abstract for this article.

Point 8: Page 7, line 229: correct the grammatical error “to promotes”.

Response 8: We thank reviewer for pointing out this error. The typo has been corrected.

Point 9: Are there studies linking specifically H2O2 induction and/or accumulation in the pathogenesis of AMD? In vivo, H2O2 is rapidly detoxified by glutathione and catalase to oxygen and water.

Response 9: The current knowledge regarding AMD pathogenesis supports that long-term exposure to accumulated lipid peroxidation products and the free radicals derived from inflammatory cells makes RPE cells vulnerable to oxidative cell death. The experimental addition of H2O2 was to mimic the microenvironment for increased external oxidative stress, but not endogenous H2O2 induction.

Round 2

Reviewer 1 Report

Unfortunately, the first 4 points of the previous round of review were not adequately answered. This is key to a quality study with rigor.

Author Response

Point 1: For the most part, this is a well-written manuscript with impressive figures. However, the manuscript lacks novelty, and if any, the data shown is incremental in nature. Perhaps authors should consider some functional studies including but not limited to phagocytosis.

Response 1: We thank for reviewer's suggestion to test additional functional experiment such as phagocytosis. Unfortunately, we have tried, but failed to establish the phagocytosis assay due to technical hindrance in preparing fresh photoreceptor outer segment fragments from porcine tissues. Instead, we provided an additional evidence showing the cytoprotective mechanism of miR-100 antagomir treatment against oxidative stress injury. Our revision data showed that miR-100 antagmir treatment significantly restored the expression of autophagosome cargo receptor p62 in ARPE-19 cells as revealed by western blotting (Fig. 6E) and immunoflorescent visualization (Fig. 7). The p62 restoration fortifies our previous hypothesis in that p62 has been previously demonstrated to activate Nrf-2 signaling by facilitating ubiquitination of its negative regulator, Keap1, from the cells and to provide cytoprotective effect on RPE cells. The revised data and possible underlying mechanism is also discussed in a newly added paragraph (Line#311). Please note that the original Fig 7 (revised Fig 8) is also modified by adding the role of p62.

Point 2: As authors have already acknowledged, the role of miR-100 in apoptosis is well documented both in cancers and in particular ocular diseases. It is unclear why RPE should be the target of miR-100, specifically in AMD. In the absence of any pre-clinical or clinical evidence that RPE is the target of miR-100, the data extrapolated from RGC’s or other cancer cell types is not justified.

Response 2: To strengthen our hypothesis for in vitro study, we have revised the Introduction text (Line#77) by adding one more reference (Reference#34), which demonstrated that the role of miR-100 expressed in retinal microvascular endothelial cells and implicates possible paracrine actions on RPE cells. In addition, we also revised the text by stating the situation of lacking pre-clinical evidence (Line#82).

Point 3: The pharmacological inhibitors used in the study are well documented to be non-specific. A more straightforward study could involve the use of dominant-negative constructs to tease out specific signaling downstream of miR-100. As presented, the data is diffuse.

Response 3:

We agree with reviewer's opinion in the possible non-specificity issue of some kinase inhibitors and thank for his/her precious suggestion. We deeply apologize for our technical inability to perform dominant-negative construct experiment. However, we rearranged one paragraph in Discussion section to put the signaling-related issues together (Line#272), in order to improve the text fluency. In that paragraph, we also added a sentence to mention the reviewer's concern on the potential non-specificity of kinase inhibitors (Line#275) and suggest that further study by using genetic intervention tools may warrant the validity of this result.

Point 4: In the absence of clear evidence that conditioned medium has extracellular vesicles that might carry micro RNA, the data shown from Fig 1C is not convincing. It is also not clear how authors have distinguished cytosolic and conditioned medium fractions and how the miR-100 expression is regulated within the cell to be secreted.

Response 4:

(1) The first question addressed by reviewer that there is no clear evidence supporting conditioned medium has extracellular vesicles that might carry micro RNA, is not true. In the first run of revision, we have added two references specifically supporting extracellular vesicles from RPE cells carry miRNA and have revised the Results text (Line#91) by adding a statement, "Recent studies have evidenced that the existence of miRNA in the extracellular vesicles released by RPE cells may contribute to retinal pathophysiolgical processes [35,36]." Moreover, we also revised Fig 1C by adding a medium control to exclude the pre-existence of miR-100 in culture medium. The statement in Methodology text is added on Line#415 accordingly.

(2) As for the second question how authors have distinguished cytosolic and conditioned medium fractions, it is undoubted that a simple high speed centrifugation could easily remove the suspension cells and debris from the conditioned medium. We therefore added a statement in the Methodology text (Line#409).

(3) We agree with reviewer that to date, it is still not clear how the miR-100 expression is regulated within the cells to be secreted. Our findings from kinase inhibition experiments suggest that PI3K/Akt, p38 and NFkappaB pathways are involved in the regulation of miR-100 expression, while PI3K/Akt and p38 signals are also related to its secretion from cultured RPE cells. We also added above statements and emphasized the significance of this and point 3 issues in the Discussion section (Line#272).

Reviewer 2 Report

After revision the manuscript improved very well and should now be considered for publication.

Author Response

We thank reviewer for his/her previously precious comments to improve the manuscript.

Round 3

Reviewer 1 Report

None